# Study of Particular Air Quality and Meteorological Parameters at a Construction Site

Lazar Milivojević [1], Sanja Mrazovac Kurilić [2,*], Zvonimir Božilović [1], Suzana Koprivica [1] and Olja Krčadinac [3]

[1] Faculty of Construction Management, Union-Nikola Tesla University, 11000 Belgrade, Serbia; lmilivojevic@unionnikolatesla.edu.rs (L.M.); zvonimirbozilovic@unionnikolatesla.edu.rs (Z.B.); skoprivica@unionnikolatesla.edu.rs (S.K.)

[2] Faculty of Ecology and Environmental Protection, Union-Nikola Tesla University, 11000 Belgrade, Serbia

[3] Faculty of Informatics, Union-Nikola Tesla University, 11000 Belgrade, Serbia; okrcadinac@unionnikolatesla.edu.rs

\* Correspondence: smrazovac@unionnikolatesla.edu.rs or mrazovac@gmail.com

**Abstract:** The construction industry is a major contributor to dust, greenhouse gases, and other air pollutants. Implementing effective and sustainable practices in managing construction site operations can greatly mitigate the environmental effects of a project. To achieve this, a collaboration between a scientific research institution and a construction company enabled the real-time monitoring of air quality parameters at a construction site using Internet of Things (IoT) technologies. They implemented an IoT-based system framework that integrated a distributed sensor network to collect real-time data from the construction site. Various sensors were utilized to gather data on the concentration of $NO_2$ and particulate matter ($PM_{2.5}$ and $PM_{10}$), as well as meteorological parameters such as wind speed, wind direction, humidity, pressure, and temperature. The real-time measurements yielded insights into the level of air pollution at the construction site and its association with earth excavation, the primary construction activity. This information can be utilized to manage excavation work and reduce the levels of polluting gases ($NO_2$) and suspended particles. By conducting an on-site monitoring of these three pollutants, the study discovered that the dust levels resulting from excavation activities were relatively high. When comparing the wind direction with $NO_2$ and PM concentrations, it was concluded that earth excavation significantly influenced the air quality in the construction area. However, in terms of the primary factors affecting $NO_2$ and construction dust concentrations, the analysis revealed that meteorological factors did not exhibit a significant correlation with $NO_2$ and dust levels at the construction site. The multiple linear regression (MLR) and the artificial neural network (ANN) models for predicting $PM_{2.5}$, $PM_{10}$ and $NO_2$ concentration in air using meteorological parameters as predictors were applied. The ANN model showed greater accordance with the measured concentrations in air than the MLR model.

**Keywords:** construction pollution; $PM_{10}$; $PM_{2.5}$; $NO_2$; meteorology; prediction model

## 1. Introduction

Air pollution is a critical issue that affects air quality worldwide, and it has significant implications for human health, ecosystems, and the environment. Air pollution is primarily caused by the emission of harmful substances into the atmosphere, including particulate matter (PM), nitrogen dioxide ($NO_2$), sulfur dioxide ($SO_2$), ozone ($O_3$), and volatile organic compounds (VOCs), among others. These pollutants are released from various sources such as industrial facilities, transportation, agriculture, and energy production.

One of the major challenges in addressing air pollution and its impacts on air quality is the lack of sufficient monitoring stations, particularly in developing countries. In many developed nations, comprehensive air quality monitoring networks are in place to measure and report pollutant concentrations accurately. However, this is not the case in numerous developing countries where the infrastructure and resources for monitoring are limited [1].

As a result, detailed information about the extent and intensity of air pollution in many areas is not adequately reported or available. This lack of data hampers the understanding of local air quality conditions, making it difficult for policymakers to implement targeted measures to tackle the problem effectively.

Furthermore, insufficient monitoring hinders the ability to establish connections between air pollution and health outcomes, which is crucial for raising awareness among the population about the risks they face and for advocating for necessary air quality regulations [2].

In conclusion, air pollution is a pressing issue that directly impacts air quality, human health, and the environment. To address air quality conditions effectively, it is vital to develop comprehensive air quality monitoring networks, especially in developing countries, to obtain accurate data and take the appropriate actions to protect public health and the environment.

The COVID-19 pandemic has had both positive and negative impacts on air pollution. The COVID-19 pandemic has revealed both short-term improvements in air quality during lockdowns and challenges in managing medical waste that indirectly affect air pollution. During the lockdowns and restrictions imposed to control the spread of the virus, there was a noticeable reduction in economic activities and transportation, leading to a temporary improvement in air quality in many urban areas. This reduction in pollution provided a glimpse of what cleaner air could look like with the appropriate measures in place [3,4].

Given the anticipated repercussions of climate change, endeavors to achieve sustainability have become paramount across various sectors, including the construction domain. Measuring the extent of atmospheric contamination resulting from diverse activities has emerged as a critical objective. Construction sites, spanning protracted periods, invariably generate substantial amounts of pollution. Within the construction industry, which accounts for approximately 12% of global emissions, the emission of greenhouse gases (GHG) remains a prominent concern. The Delhi Pollution Control Committee (DPCC) has officially reported that emissions from construction sites account for 30% of dust-related air pollution. Excavation, the operation of diesel engines, demolition, incineration, and the handling of toxic substances are among the manifold construction undertakings that contribute to air pollution. The principal catalyst behind the release of nitrogen and sulfur oxides during construction projects is the utilization of heavy machinery—namely, excavators, loaders, bulldozers, and others—which burn fossil fuels. Excavation work primarily attributes to particulate matter (PM) pollution on construction sites. Diesel engine exhausts, diesel generator sets, vehicles, and heavy equipment constitute significant sources of $PM_{2.5}$. Moreover, the air pollution dilemma is exacerbated by the emission of harmful substances from oils, adhesives, solvents, paints, treated timber, plastics, cleaning agents, and other perilous chemicals widely employed within construction sites [5].

In 2015, the Sustainable Development Goals (SDGs), also known as the 2030 Agenda for Sustainable Development, were adopted by 193 countries [6]. Within these goals, air pollution is specifically addressed in two targets: SDG 3.9, which aims for a substantial reduction in the health impacts from hazardous substances, and SDG 11.6, which focuses on reducing the adverse effects of cities on people. Taking action in the energy sector is crucial for achieving these SDGs related to air pollution [7,8]. The majority of sulfur dioxide ($SO_2$) and nitrogen oxide ($NO_X$) emissions, as well as approximately 85% of particulate matter (PM) emissions, are associated with energy-related activities. These three primary pollutants have significant direct and indirect consequences on air pollution, resulting from chemical reactions and atmospheric transport. Among them, $PM_{2.5}$ poses the highest risk to human health, while sulfur and $NO_X$ (which contribute to ozone formation) are linked to various illnesses and environmental harm [5,9]. The Sustainable Development Scenario (SDS) is designed to align with the selected SDGs established by the United Nations. It aims to achieve three interrelated objectives: ensuring universal access to affordable, reliable, and modern energy services by 2030 (SDG 7.1), significantly reducing air pollution that leads to high mortality and disease (SDG 3.9) and taking effective measures to combat climate

change (SDG 13). Serbia leads the construction sector in the Balkan region, experiencing annual growth. In August 2022, there were 2562 permits granted for construction projects. This upward trend in construction poses the risk of significantly increasing greenhouse gas concentrations and other pollutants. Consequently, it becomes imperative to establish the real-time monitoring of noxious gases and particulate matter (PM). Such monitoring aims to provide insights into pollutant levels and their correlation with atmospheric conditions, thereby facilitating the formulation of measures to mitigate the concentration of harmful emissions.

Despite the escalating construction activities in Serbia, the implementation of a real-time emission monitoring tool remains absent from construction sites. This tool is crucial in helping construction teams prevent the excessive release of harmful substances. The significance of adopting such a system and conducting this type of research lies in safeguarding the health of construction site workers, who often face health issues due to adverse working conditions and poor air quality. At times, the air quality at these sites deteriorates to such an extent that it poses a direct threat to the lives of the labor force.

This problem deserves much more attention due to the impact on the health of the population in the immediate vicinity of the construction site, too. The results of this research provide the new possibility of predicting air quality at the construction site, given the possibility of predicting meteorological parameters. This fact is significant because it suggests the possibility of planning and managing the works in such a way that the health of the workers, the population, and the environment are minimally affected. This way of working and this approach to complex activities on the construction site represents a sustainable way of managing the construction site and the construction project as a whole.

Particulate matter (PM) is among the most pervasive air pollutants globally, alongside $NO_X$, photochemical oxidants, ozone ($O_3$), carbon monoxide (CO), lead (Pb), and sulfur dioxide ($SO_2$) [10,11]. Recent studies have focused on dust concentration at construction sites, specifically examining $PM_{10}$ and $PM_{2.5}$ [12–14]. These studies have identified numerous factors influencing PM concentrations at construction sites. Notably, the surrounding areas of the construction site can serve as a source of emissions that are transported and detected at the site, independent of on-site activities, referred to as background emissions. The impact of meteorological factors on pollutant concentrations, including PM, has been explored through several investigations, yielding conflicting perspectives. Certain authors [15] emphasize the considerable influence of meteorological parameters on PM concentrations at construction sites, yet due to limited measured data, a model linking PM concentrations to meteorological parameters remains elusive. Conversely, other researchers [16] contend that dust emissions from construction sites exhibit significant seasonal fluctuations, corroborated by additional studies [17]. This underscores the strong correlation between PM concentration and meteorological parameters. Some research [18,19] exploring the relationship between construction activities and meteorological parameters has found a highly positive correlation between PM and wind speed and relative humidity, while the correlation with temperature is weak. Apart from excavation work, internal construction activities within buildings also contribute to emissions. Kinsey et al. [20] discovered that vehicles departing from construction sites can carry substantial amounts of dust and sediment onto nearby roads, resulting in secondary dust dispersion. Azarmi et al. [21] carried out extensive monitoring during specific stages of work, including concrete mixing, drilling, and cutting. They found that concentrations of $PM_{10}$, $PM_{2.5}$, and $PM_{0.1}$ during drilling and cutting activities were up to 14 times higher than the background levels [21]. In a study by Moraes et al. [12], the focus was on monitoring $PM_{10}$ concentrations resulting from concrete and masonry work in construction activities. These and similar investigations have provided evidence that certain work phases and activities on construction sites play a crucial role in influencing PM concentrations [22].

The objective of this research is to conduct a more comprehensive and detailed analysis of the relationship between $NO_2$ and PM concentrations emitted from excavation work on construction sites and meteorological parameters. The data analysis aims to explore the

feasibility of employing predictive models to estimate $NO_2$ and PM concentrations based on meteorological parameters.

## 2. Materials and Methods

The investigation encompassed the assessment of airborne concentrations of $PM_{2.5}$, $PM_{10}$, and $NO_2$, as well as the measurement of meteorological variables (including air pressure, temperature, humidity, wind speed, and wind direction). The study was conducted at a construction site in Belgrade (refer to Figure 1) over a 15-day period in July 2022, spanning from the first to the fifteenth of the month. Notably, the excavation zone is situated to the west of the monitoring station, while other emission sources on the construction site, such as the construction waste disposal area, carpentry workshop, and reinforcement work, are located north of the monitoring device. Figure 2 illustrates the distances between the measuring station and each individual emission source. Throughout the weekdays (Monday to Saturday), two electric-powered machines (excavators) were employed in the excavation zone. Heavy excavation activities were conducted from 07:00 to 17:00 h, except on Sundays.

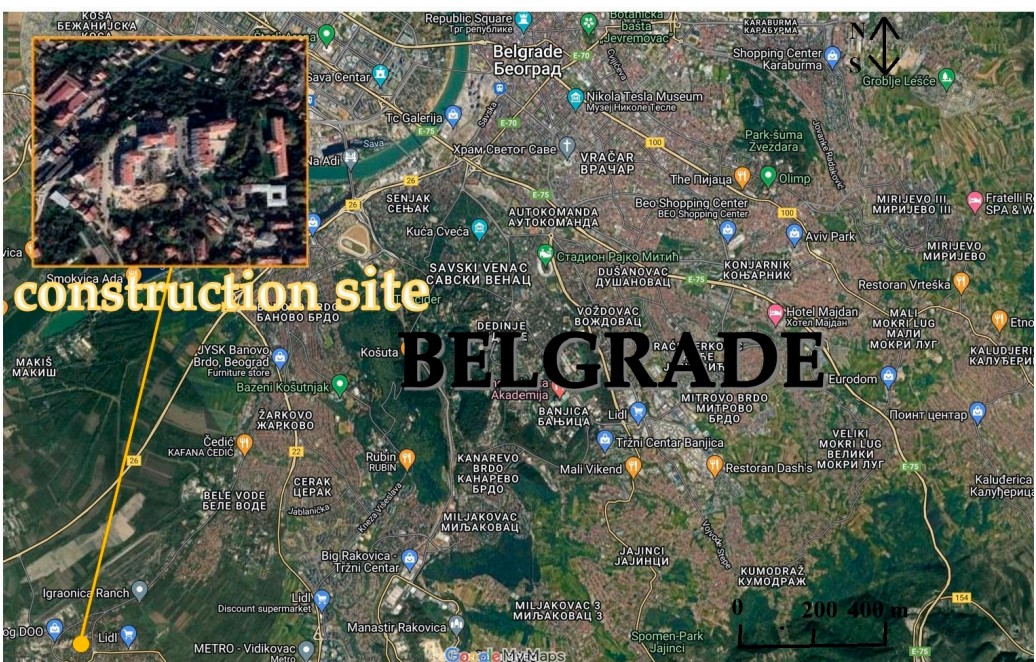

**Figure 1.** Location of the construction site in Belgrade, Serbia.

The waste excavated material was transported off-site once every day by the same truck. It was not possible, and so was not a goal of this study, to reliably confirm the origins of the polluting substances in the air, but the main off-site factor affecting the real-time data on the air pollutant concentration collected near the construction site is traffic, specifically several busy road routes located at relatively short distances from the construction site, as shown in Figure 1. However, during the entire measurement period, heavy earth excavation works were carried out and were temporally predominant, while the only other work, interior construction inside nearby buildings, was only very sporadically conducted during the measurement period. For this reason, we believe that we can attribute the air pollution at the site to the heavy earthworks. Therefore, the focus of this study was on the impact of meteorological conditions on PM and $NO_2$, considered to be from the heavy earth excavation works, in the air.

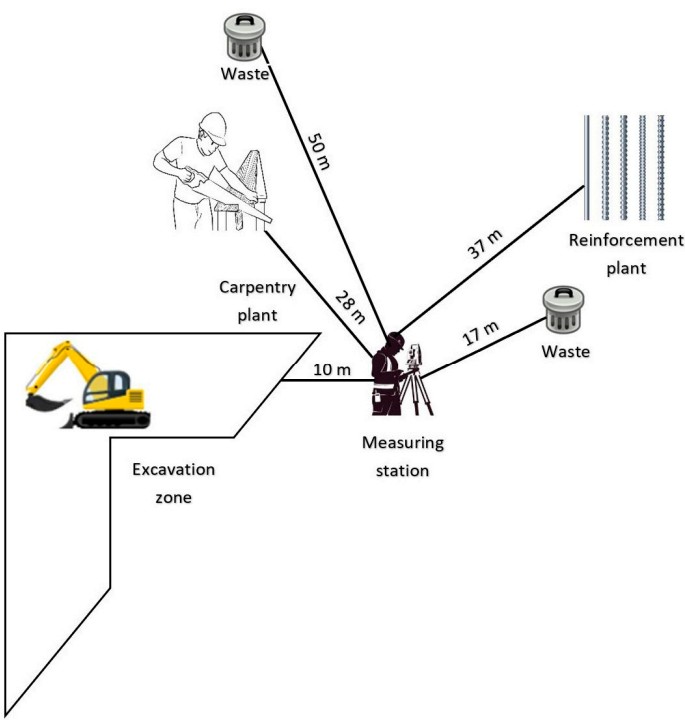

**Figure 2.** Illustration of the construction site indicating the locations of the monitoring station, excavation area, and other possible emission sources.

The measurement devices utilized were portable sensors capable of indoor and outdoor use. Housed within the measurement station, these devices recorded measurements every 5 min. The RS-MG111-WIFI-1 (Shandong Renke Control Technology Co., Ltd, Shandong, China) served as an air environment multi-element transmitter, detecting $NO_2$, $PM_{2.5}$, and $PM_{10}$ concentrations at the measurement site (Figure 3). Equipped with an imported sensor and a control chip known for its high precision, resolution, and stability, this transmitter seamlessly connected to the on-site WIFI network. It formed an integrated online air environment monitoring system, commonly employed in various settings such as smart homes, schools, hospitals, airports, and train stations, offering energy-saving solutions in heating, ventilation, and air conditioning systems. Another device used was the CC-M12 weather station with RH&T and 4G communication, which measured wind direction (WD), wind speed (WS), air temperature, air pressure, and humidity (Chao Sensor Group, Zhejiang, China) (Figure 4).

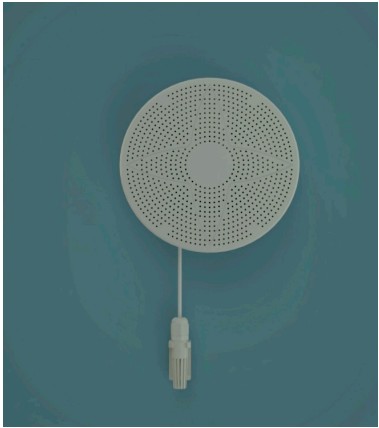

**Figure 3.** RS-MG111-WIFI-1 (Reinke) device.

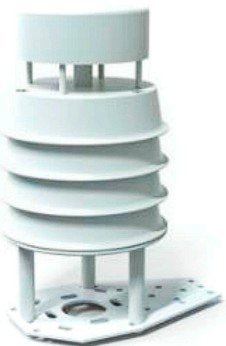

**Figure 4.** CC-M12 weather station.

This comprehensive system provided real-time insight into air quality for construction site managers and company stakeholders. It facilitated the identification of harmful gas emissions from three primary construction activities: earthworks, transportation, and interior works. Web and mobile applications enabled data visualization through maps, lists, and charts, along with notifications and alarms for values exceeding predefined thresholds. The system incorporated algorithms for data processing and allowed for data export in CSV format.

The price and the parameters that can be measured were significant factors when choosing a multi-sensor. The sensor is positioned in the middle part of the construction site, surrounded by construction activities at a relatively short distance, enabling the detection of polluting substances resulting from those activities. In our research, we utilized a combination of optimization techniques to ensure accurate and reliable measurements of PM particles and $NO_2$ levels. The sensors' calibration was carefully conducted to minimize any potential biases and enhance their precision. To assess the sensors' performance, we conducted rigorous comparative analyses with reference-grade instruments known for their accuracy and commonly used as the standard in air quality monitoring. This comparison allowed us to evaluate the performance of our IoT sensors and identify any potential discrepancies or variations in the measurements. Moreover, we addressed the validation of the IoT sensors by conducting field tests in diverse environmental conditions. This validation process involved monitoring air quality at different locations and times, covering various pollutant levels and weather conditions. The collected data were then compared against established air quality indices and regulatory standards to ensure the reliability and credibility of our sensor measurements.

To ensure data quality, the sensors were calibrated against an official site, demonstrating an accuracy exceeding 0.98. Calibration involved a field collocation method, where a low-cost device was colocated with a public air quality monitoring station for a 15-day period, capturing hourly averaged values from both devices. A Least Squares Method (LSM) [23] was employed as one of the commonly used calibration techniques due to its simplicity.

A research flow chart presents the research process (Figure 5). For data analysis in this study, SPSS 23.0 statistical software and Excel were utilized. Data modeling, including multiple linear regression (MLR) and artificial neural network (ANN), was performed using Statistica v.13 software (StatSoft, Dell, Round Rock, TX, USA).

The MLR model involved fitting a linear equation to the observed data, allowing for the examination of relationships between variables without indicating a causal mechanism. This model was crucial in determining how meteorological factors influenced air pollutant concentrations. Consequently, $NO_2$ and PM concentrations were treated as responses to meteorological variables acting as predictors.

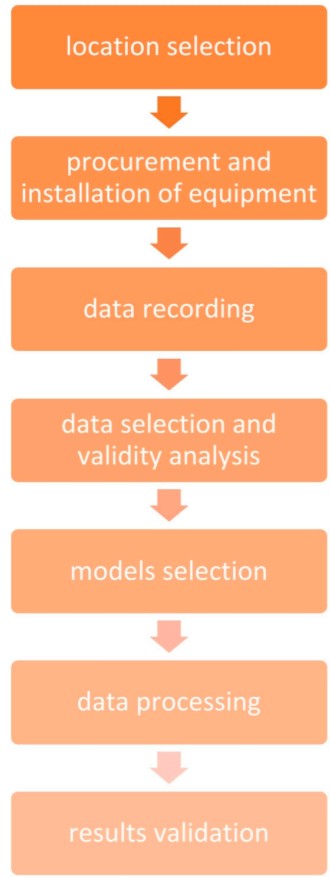

**Figure 5.** Research flow chart.

The ANN, a widely used prognostic method, served as a solution when other statistical techniques were not applicable. Its advantages, such as learning from examples, fault tolerance, real-time operation, and forecasting non-linear data, made it the preferred choice. The ANN models accurately captured non-linear variables, distinguishing themselves from the multivariate linear analysis that relies on linear variables. These models aimed to simulate the functioning of neurons in the human brain through mathematical functions. The multilayer perception (MLP) consisted of input layers corresponding to input data, hidden layers with interconnected artificial neurons, and an output layer with "target" neurons for predictions.

The coefficient of determination ($R^2$) was employed as an indicator to assess whether the data provided sufficient evidence for reliable predictions. It measured the degree to which the prediction models fit the data, with values ranging from zero to one. A higher value closer to one indicated a more accurate prediction.

### 3. Results and Discussion

The measurement results over the measurement period are shown in Figures 6 and 7. Data are given for work hours, showing the separate work hours from 07:00 to 17:00 h on work days (Monday to Saturday). Three sets of data were obtained by monitoring the concentrations of polluting substances ($PM_{2.5}$, $PM_{10}$, and $NO_2$) in air. As shown in Figures 3 and 4, these three sets of data are plotted as box plots ($NO_2$, $PM_{10}$, and $PM_{2.5}$). From the results shown, it can be seen that $PM_{2.5}$ concentrations ranged from 1 to 38 $\mu g/m^3$. The mean $PM_{2.5}$ concentration during work hours was 14.66 $\mu g/m^3$. The 24-h mean concentrations of $PM_{2.5}$ for all 15 days were: 26.46, 14.69, 21.06, 26.87, 27.09, 15.76, 15.16, 16.55, 11.66, 7.26, 5.75, 9.38, 8.36, 10.20, and 15.26 $\mu g/m^3$.

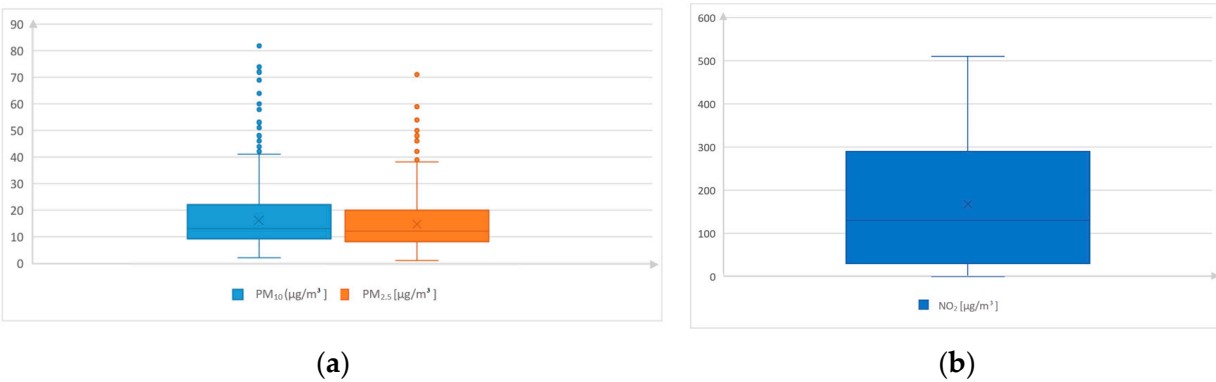

**(a)**            **(b)**

**Figure 6.** (**a**) Mean concentration of $PM_{10}$ and $PM_{2.5}$ in the air (minimum, 1st quartile, median and mean value, 3rd quartile, and maximum, as well as outliers are shown) at the construction site during excavation works in a period of 15 days (data from work hours). (**b**) Mean concentration of $NO_2$ in the air at the construction site during excavation works in a period of 15 days (data from work hours).

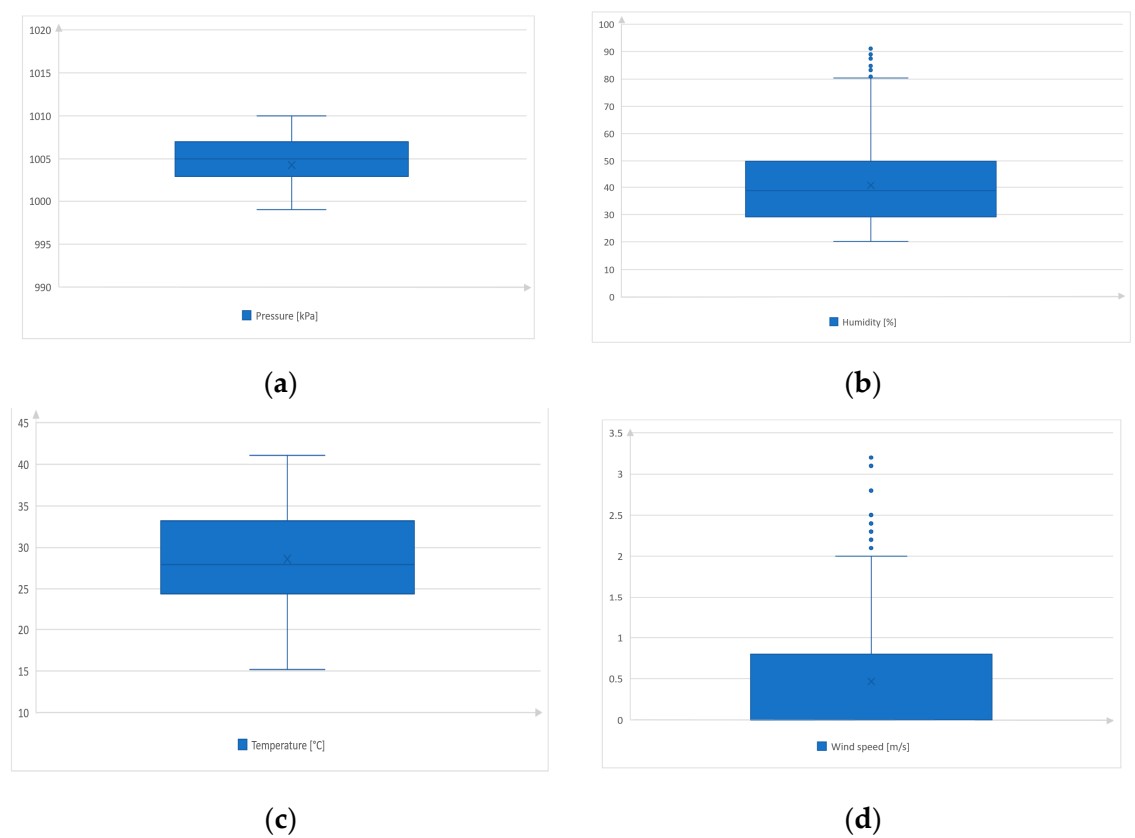

**(c)**            **(d)**

**Figure 7.** (**a**) Atmospheric pressure at the construction site during excavation works in a period of 15 days (data from work hours). (**b**) Humidity at the construction site during excavation works in a period of 15 days (data from work hours). (**c**) Temperature at the construction site during excavation works in a period of 15 days (data from work hours). (**d**) Wind speed at the construction site during excavation works in a period of 15 days (data from work hours).

$PM_{10}$ concentrations during work hours ranged from 2 to 41 $\mu g/m^3$. The mean concentration was 16.05 $\mu g/m^3$. The 24-h mean concentrations of $PM_{10}$ for all 15 days were: 29.18, 16.22, 23.05, 30.21, 30.15, 16.97, 16.04, 17.50, 12.69, 7.98, 6.48, 10.55, 9.11, 11.08, and 16.94 $\mu g/m^3$. The highest $PM_{10}$ and $PM_{2.5}$ concentrations were measured during the night (non-work) hours, which could be attributed to the stable stratification of the atmosphere, according to [24].

The relationship between $PM_{10}$ and $PM_{2.5}$ concentrations (from a daily average) was calculated. The $PM_{2.5}$ concentration was approximately 90% of the $PM_{10}$ concentration. The $PM_{2.5}$ to $PM_{10}$ ratios for all 15 days were: 90.7%, 90.6%, 91.4%, 88.9%, 89.8%, 92.9%, 94.5%, 94.6%, 91.9%, 91.0%, 88.7%, 88.9%, 91.8%, 92.1%, and 90.1%. In accordance with the calculated values, we can indirectly estimate the emission sources. High ratios indicate industrial and traffic emissions.

Based on the limits set by the World Health Organization (WHO), Geneva, Switzerland [25], the annual mean for $PM_{2.5}$ concentration should not surpass 5 $\mu g/m^3$, while the 24-h mean should not exceed 15 $\mu g/m^3$. For $PM_{10}$, the annual mean limit is 15 $\mu g/m^3$, and the 24-h mean limit is 45 $\mu g/m^3$. After analyzing the average 24-h means of $PM_{2.5}$ and $PM_{10}$ concentrations at our construction site, it can be concluded that $PM_{2.5}$ poses a significantly higher health risk. This is due to the measured concentrations consistently exceeding the prescribed daily limit set by the WHO, even on non-work days, on more than 50% of occasions. In contrast, the $PM_{10}$ concentrations did not exceed the permissible 24-h mean, according to WHO standards, as frequently. The Republic of Serbia implemented the Law on Air Protection in 2009 to align with European Union (EU) regulations, which have less stringent standards compared to the WHO. Both $PM_{2.5}$ and $PM_{10}$ concentrations display a right-skewed distribution pattern.

The $NO_2$ concentrations (during work hours) ranged from 0 to 510 $\mu g/m^3$. The mean $NO_2$ concentration during work hours was 167.741 $\mu g/m^3$. A significantly higher $NO_2$ concentration was observed at the construction site during work hours than during non-work hours. For the entire 15 days, the work-hours mean $NO_2$ concentration was about 70% higher than the 24-h mean. The concentration of $NO_2$ in the air could have been impacted by the transport of the waste that was taken to the construction waste disposal site by truck every day, but also by the off-site traffic from nearby roads. For $NO_2$ a right-skewed distribution can be observed too.

Four sets of meteorological data (wind speed, temperature, humidity, and atmospheric pressure) were obtained. As shown in Figure 7, the four sets of data are plotted in box plots. The pressure ranged from 999 to 1010 kPa throughout the work hours. The average atmospheric pressure was 1004.307 kPa. For pressure, a left-skewed distribution can be observed.

The work-hours mean humidity ranged from 13.75 to 80.4%, and the work-hours mean humidity was 40.67%.

The work-hours mean air temperature ranged from 15.2 to 41.1 °C. The work-hours mean air temperature was 28.6 °C. For humidity and temperature, right-skewed distributions can be observed.

The wind speed (24-h mean) ranged from 0 to 2 m/s during work hours. The mean wind speed during work hours was 0.467 m/s.

Table 1 shows the values of the Spearman correlation coefficients for the measured parameters, from which we can conclude that the concentrations of $PM_{10}$ and $PM_{2.5}$ were not significantly correlated with any meteorological factor. A very high correlation between $PM_{2.5}$ and $PM_{10}$ concentrations was observed. This coefficient was chosen in accordance with the fact that the data do not have a normal distribution.

The absence of a correlation between dust and the examined meteorological factors can be attributed to the multifaceted nature of construction dust influences. Construction activities directly contribute to the generation of construction dust, making them the primary driver [20] of construction dust levels, surpassing the impact of meteorological factors. Throughout the monitoring period, the meteorological conditions remained relatively stable, which may have mitigated or limited the influence of meteorological factors on construction dust. Precipitation emerges as the primary factor that affects dust levels. Consequently, it can be inferred that the emission of construction dust shows no significant association with any meteorological factor when these factors exhibit minimal variation. To some extent, this finding aligns with the conclusions drawn from studies investigating urban $PM_{10}$ and $PM_{2.5}$ levels [26,27].

**Table 1.** Values of the Spearman correlation coefficient among the measured parameters.

| | NO$_2$ | PM$_{10}$ | PM$_{2.5}$ | Pressure | Humidity | Temperature | Wind Direction | Wind Speed |
|---|---|---|---|---|---|---|---|---|
| NO$_2$ | 1 | 0.239763 | 0.253295 | 0.051102 | −0.37261 | 0.957658 | 0.333358 | 0.269198 |
| PM$_{10}$ | | 1 | 0.986733 | −0.07757 | 0.165151 | 0.194665 | 0.177881 | 0.14115 |
| PM$_{2.5}$ | | | 1 | −0.10016 | 0.149039 | 0.2059 | 0.17116 | 0.134286 |
| Pressure | | | | 1 | 0.763782 | −0.02965 | 0.576633 | 0.668878 |
| Humidity | | | | | 1 | −0.44567 | 0.402222 | 0.520197 |
| Temperature | | | | | | 1 | 0.271544 | 0.195947 |
| Wind direction | | | | | | | 1 | 0.819766 |
| Wind speed | | | | | | | | 1 |

Figure 8 shows pollution roses for NO$_2$, PM$_{2.5}$, and PM$_{10}$, indicating the relationship between the pollutant concentrations and the wind direction. Clearly, the S-W wind was the predominant wind direction as it had the greatest contribution to the concentrations of air pollutants (Figure 2). This was likely due to the nearby excavation zone.

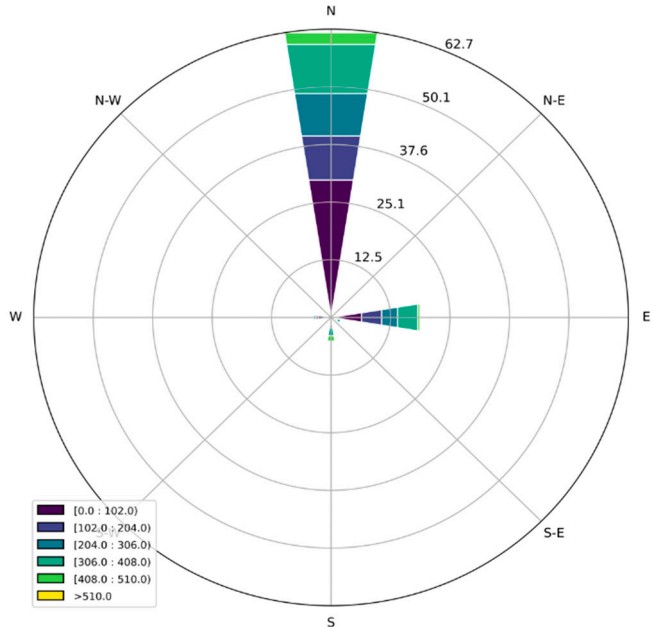

(a)

**Figure 8.** *Cont.*

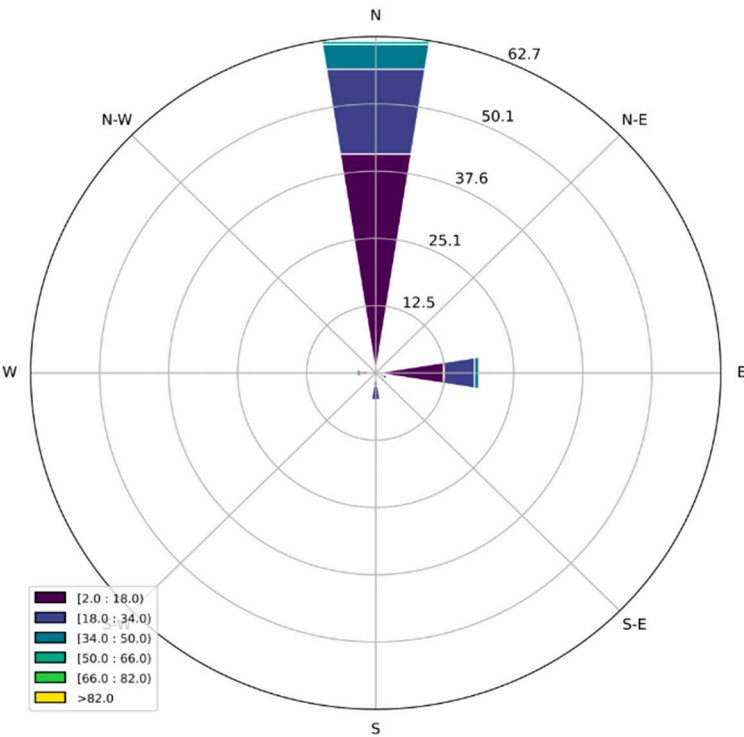

(b)

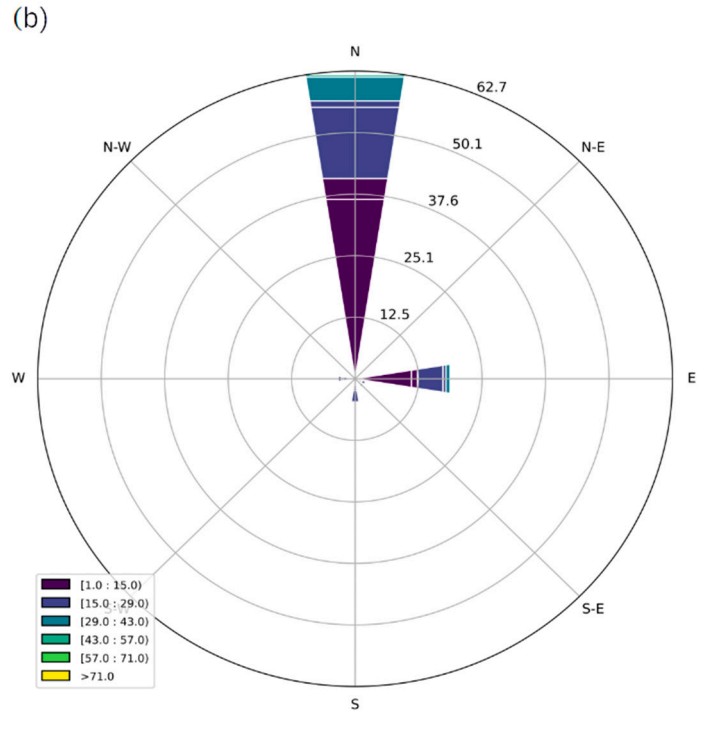

(c)

**Figure 8.** Pollution roses for NO$_2$, PM$_{10}$, and PM$_{2.5}$ (concentrations in μg/m$^3$). (**a**) Pollution rose for NO$_2$. (**b**) Pollution rose for PM$_{10}$. (**c**) Pollution rose for PM$_{2.5}$.

The Conditional Probability Function (CPF) calculates the probability that a source is located within a particular wind direction sector (Figure 9). CPF is useful for determining the direction of a source with respect to a receptor site. However, it cannot determine the

actual location of the source. Pollution in this case, for $NO_2$ concentrations, comes from the south, predominantly (Figure 9a).

PM$_{10}$ pollution comes from the south, southwest and west, predominantly (Figure 9b). The location of the excavation zone is west of the measuring station, so the real-time data on the air pollutant concentration collected cannot all be attributed to the earthworks. Similar results are shown for PM$_{2.5}$ (Figure 9c).

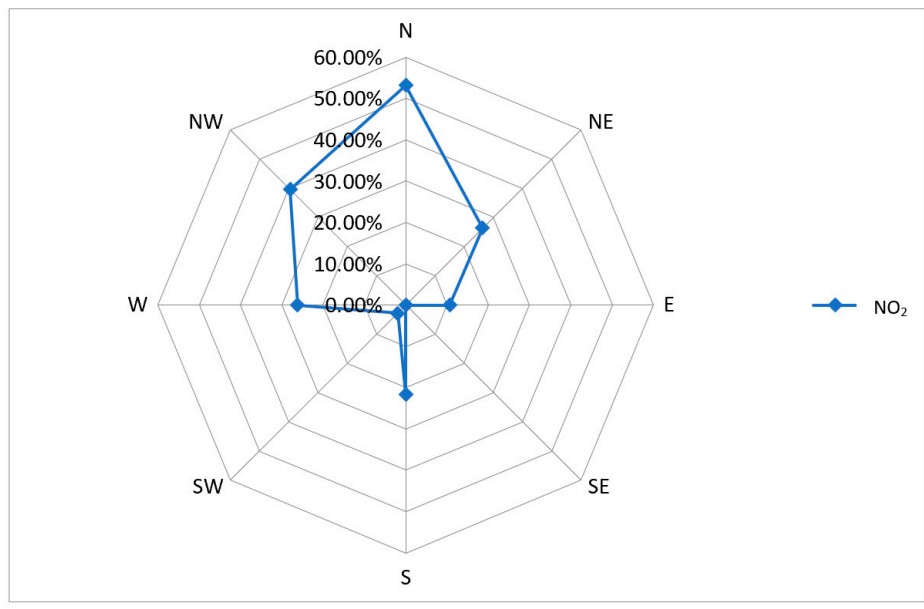

(**a**)

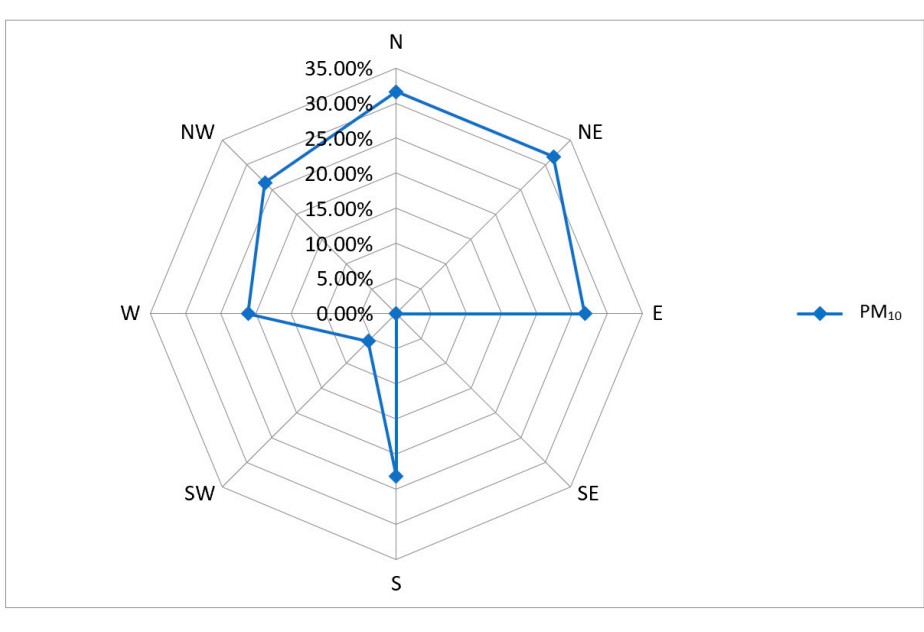

(**b**)

**Figure 9.** *Cont.*

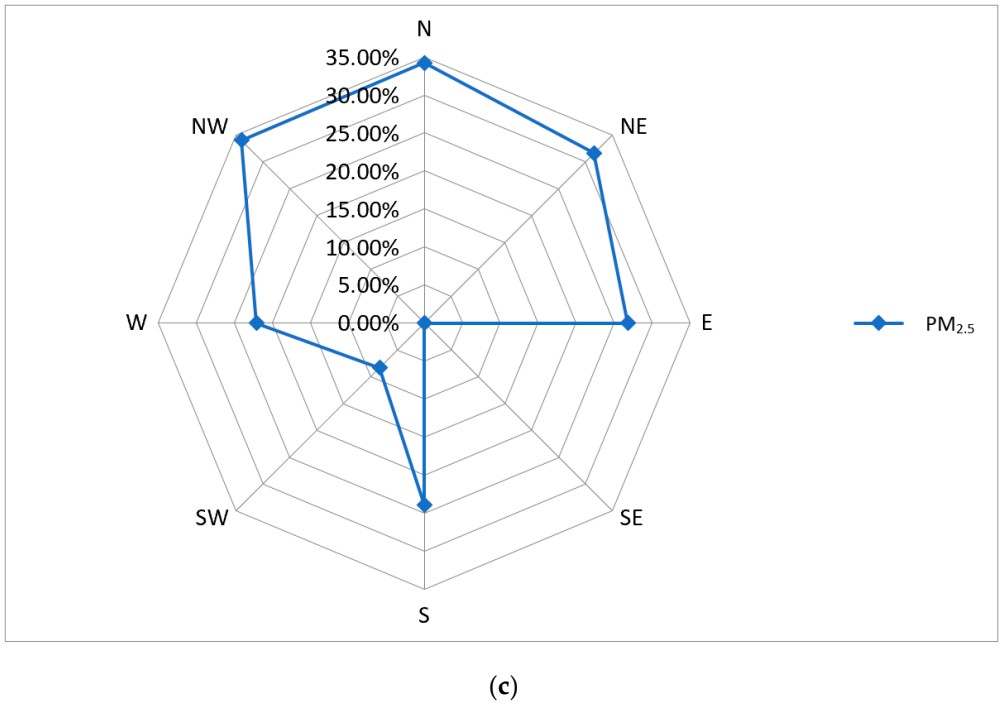

(**c**)

**Figure 9.** CPF for $NO_2$, $PM_{10}$ and $PM_{2.5}$ concentrations above 75th percentile. (**a**) CPF for $NO_2$ concentrations above 75th percentile. (**b**) CPF for $PM_{10}$ concentrations above 75th percentile. (**c**) CPF for $PM_{2.5}$ concentrations above 75th percentile.

Two prediction models were developed based on the experimental data, the MLR (multiple linear regression) and the ANN (artificial neural network) models (Figure 10). A set of 1492 data were used, since not all of the expected 1800 data were valid. In shorter periods of time during the operation of the device for measuring parameters, the power supply was interrupted, or the internet connection was interrupted.

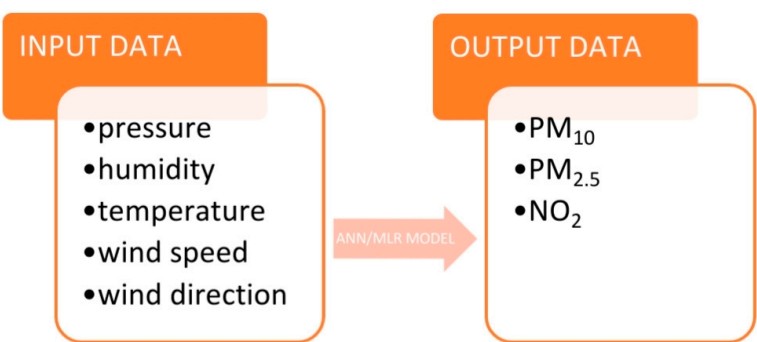

**Figure 10.** Input and output data for ANN and MLR models.

### 3.1. Prediction Model for Air Pollutant Concentrations: ANN-Model

For the ANN model, 1040 data were used for training, 452 for testing, and 1492 for model validation. Predictors for the model for $NO_2$, $PM_{10}$ and $PM_{2.5}$ prediction were wind speed (m/s), pressure (kPa), humidity (%), wind direction (°), and temperature (°C). The dependent variables were the concentrations of $NO_2$ ($\mu g/m^3$), $PM_{10}$ ($\mu g/m^3$) and $PM_{2.5}$ ($\mu g/m^3$). The $R^2$ coefficient of determination for $NO_2$ was 0.967, for $PM_{10}$ it was 0.668, and for $PM_{2.5}$ it was 0.706. The results are presented in Figure 11.

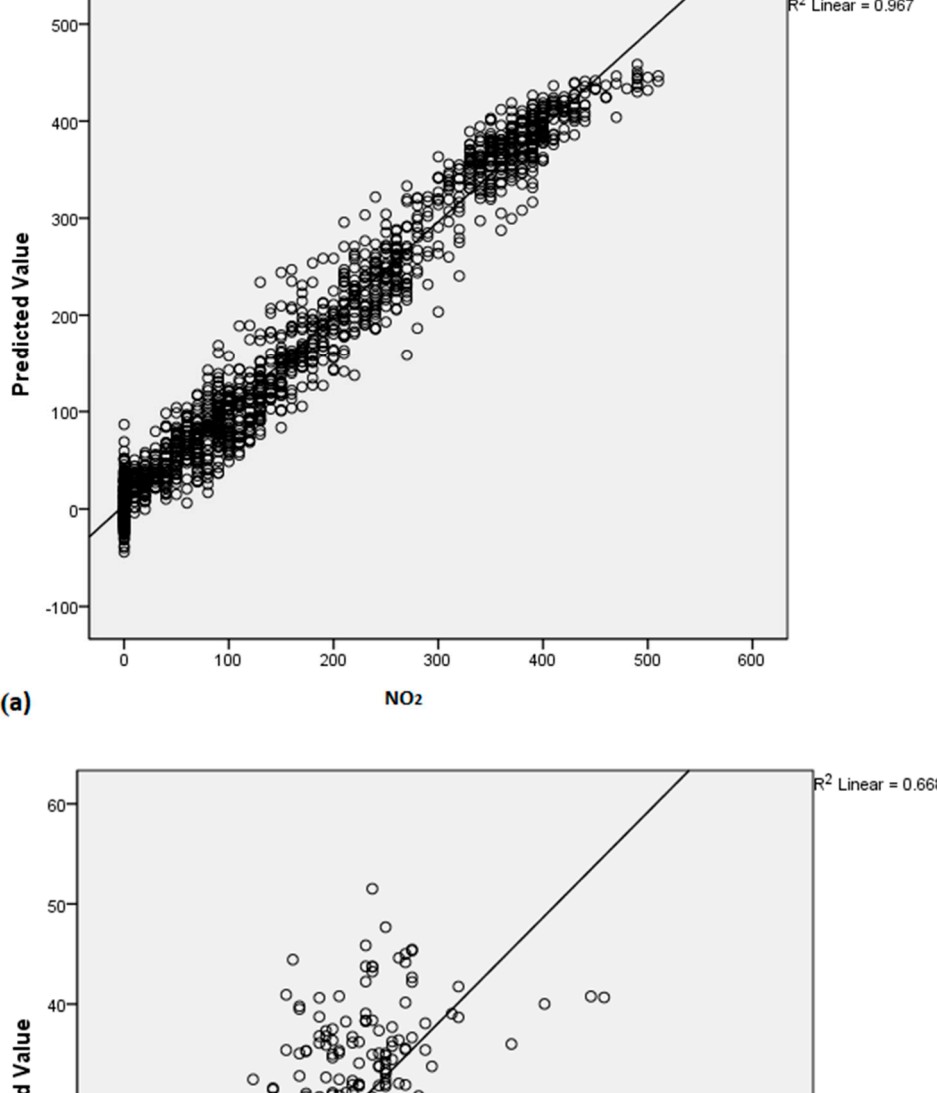

**Figure 11.** *Cont.*

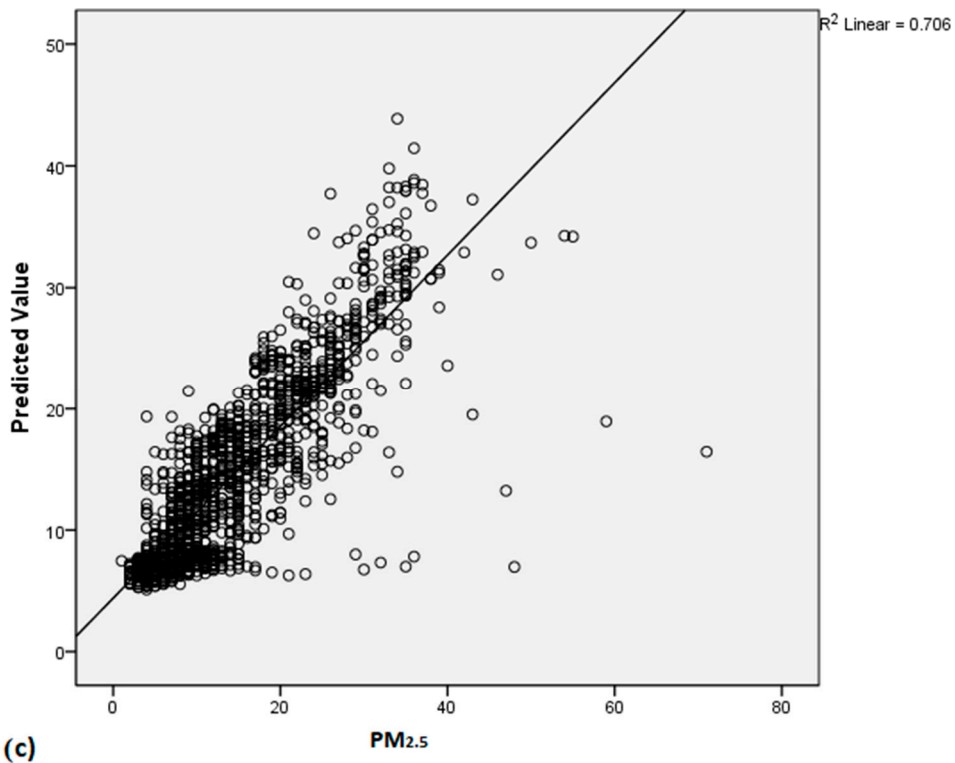

**Figure 11.** ANN model results for (**a**) NO$_2$, (**b**) PM$_{10}$, and (**c**) PM$_{2.5}$ concentrations in μg/m$^3$.

### *3.2. Prediction Model for Air Pollutant Concentrations: MLR Model*

The predictors for the model for NO$_2$ prediction were wind speed (m/s), pressure (kPa), humidity (%), wind direction (°), and temperature (°C). The dependent variable was NO$_2$ (μg/m$^3$). Model results for NO$_2$ are shown in Tables 2–4.

**Table 2.** Model summary (R—coefficient of correlation, R$^2$—coefficient of determination, Std. Error of the Estimate—standard error of the estimate) for NO$_2$ prediction.

| R | R$^2$ | Adjusted R$^2$ | Std. Error of the Estimate |
|---|---|---|---|
| 0.958 | 0.917 | 0.917 | 41.700 |

**Table 3.** ANOVA table (Df—degrees of freedom, F—ratio of between group variation and within group variation, Sig.—significance level) for NO$_2$ prediction.

| Model | Sum of Squared | Df | Mean Square | F | Sig. |
|---|---|---|---|---|---|
| Regression | 28,724,409.308 | 5 | 5,744,881.862 | 3303.797 | 0.000 |
| Residual | 2,583,964.619 | 1486 | 1738.873 | | |
| Total | 31,308,373.928 | 1491 | | | |

The predictors for the model for PM$_{10}$ prediction were wind speed (m/s), pressure (kPa), humidity (%), wind direction (°), and temperature (°C). The dependent variable was PM$_{10}$ concentration (μg/m$^3$). Model results for PM$_{10}$ are shown in Tables 5–7.

**Table 4.** Model coefficients (t—t-statistics, Sig.—significance) for $NO_2$ prediction.

| Model | Unstandardized Coefficients | | Standardized Coefficients | t | Sig. |
|---|---|---|---|---|---|
| | **B** | **Std. Error** | **Beta** | | |
| Constant | −899.221 | 633.401 | | −1.420 | 0.156 |
| Pressure (kPa) | 0.380 | 0.618 | 0.007 | 0.615 | 0.539 |
| Humidity (%) | −0.361 | 0.129 | −0.034 | −2.791 | 0.005 |
| Temperature (°C) | 24.464 | 0.429 | 0.938 | 57.070 | 0.000 |
| Wind direction (°) | 0.017 | 0.018 | 0.008 | 0.949 | 0.343 |
| Wind speed (m/s) | −1.368 | 2.227 | −0.005 | −0.614 | 0.539 |

**Table 5.** Model summary (R—coefficient of correlation, $R^2$—coefficient of determination, Std. Error of the Estimate—standard error of the estimate) for $PM_{10}$ prediction.

| R | $R^2$ | Adjusted $R^2$ | Std. Error of the Estimate |
|---|---|---|---|
| 0.726 | 0.527 | 0.525 | 7.290 |

**Table 6.** ANOVA table (Df—degrees of freedom, F—ratio of between group variation and within group variation, Sig.—significance level) for $PM_{10}$ prediction.

| Model | Sum of Squares | df | Mean Square | F | Sig. |
|---|---|---|---|---|---|
| Regression | 87,955.640 | 5 | 17,591.128 | 331.014 | 0.000 |
| Residual | 78,970.743 | 1486 | 53.143 | | |
| Total | 166,926.383 | 1491 | | | |

**Table 7.** Model coefficients (t—t-statistics, Sig.—significance) for $PM_{10}$ prediction.

| Model | Unstandardized Coefficients | | Standardized Coefficients | t | Sig. |
|---|---|---|---|---|---|
| | **B** | **Std. Error** | **Beta** | | |
| Constant | 1948.335 | 110.731 | | 17.595 | 0.000 |
| Pressure (kPa) | −1.951 | 0.108 | −0.523 | −18.068 | 0.000 |
| Humidity (%) | 0.418 | 0.023 | 0.543 | 18.479 | 0.000 |
| Temperature (°C) | 0.432 | 0.075 | 0.227 | 5.771 | 0.000 |
| Wind direction (°) | 0.007 | 0.003 | 0.048 | 2.276 | 0.023 |
| Wind speed (m/s) | −1.971 | 0.389 | −0.107 | −5.063 | 0.000 |

The predictors for the model for $PM_{2.5}$ prediction were wind speed (m/s), pressure (kPa), humidity (%), wind direction (°), and temperature (°C). The dependent variable was $PM_{2.5}$ concentration ($\mu g/m^3$). Model results for $PM_{2.5}$ are shown in Tables 8–10.

**Table 8.** Model summary (R—coefficient of correlation, $R^2$—coefficient of determination, Std. Error of the Estimate—standard error of the estimate) for $PM_{2.5}$ prediction.

| R | $R^2$ | Adjusted $R^2$ | Std. Error of the Estimate |
|---|---|---|---|
| 0.765 | 0.586 | 0.584 | 5.900 |

**Table 9.** ANOVA table (Df—degrees of freedom, F—ratio of between group variation and within group variation, Sig.—significance level) for $PM_{2.5}$ prediction.

| Model | Sum of Squares | Df | Mean Square | F | Sig. |
|---|---|---|---|---|---|
| Regression | 73,093.861 | 5 | 14,618.772 | 419.966 | 0.000 |
| Residual | 51,726.752 | 1486 | 34.809 | | |
| Total | 124,820.613 | 1491 | | | |

**Table 10.** Model coefficients (t—t statistics, Sig.—significance) for $PM_{2.5}$ prediction.

| Model | Unstandardized Coefficients | | Standardized Coefficients | t | Sig. |
|---|---|---|---|---|---|
| | B | Std. Error | Beta | | |
| Constant | 1730.920 | 89.617 | | 19.315 | 0.000 |
| Pressure (kPa) | −1.736 | 0.087 | −0.538 | −19.858 | 0.000 |
| Humidity (%) | 0.384 | 0.018 | 0.577 | 20.989 | 0.000 |
| Temperature (°C) | 0.453 | 0.061 | 0.275 | 7.477 | 0.000 |
| Wind direction (°) | 0.006 | 0.002 | 0.048 | 2.409 | 0.016 |
| Wind speed (m/s) | −1.570 | 0.315 | −0.099 | −4.984 | 0.000 |

From the applied models (Tables 2–10) and Figure 11, based on $R^2$ values, the ANN model showed greater accordance with the measured air pollutant concentrations than the MLR model.

## 4. Conclusions

The main motivation of the presented research is to investigate the impact of construction activities on air quality at the construction site itself, directly affecting the health of the workers and the immediate environment. Additionally, the challenge lies effectively in organizing activities to minimize harmful impacts. By aligning activities with weather conditions, to the greatest extent possible, we can significantly improve environmental quality. Therefore, it is crucial to establish a connection between weather conditions, meteorological parameters, and air quality parameters. Understanding their interrelation can aid in predicting pollution levels based on forecasted weather conditions, providing valuable information for sustainable construction management.

To assess the primary factors influencing dust and $NO_2$ concentrations originating from construction activities and their impact on the construction site, meteorological data and airborne pollutant measurements were collected. The findings aimed to establish a foundation for mitigating the effects of construction-related dust and $NO_2$ emissions on the construction area. The monitoring conducted at a construction site in Belgrade City revealed significantly elevated dust concentrations during construction activities. The average $PM_{10}$ concentration during working hours was 16.05 $\mu g/m^3$, while the average $PM_{2.5}$ concentration was 14.7 $\mu g/m^3$. Additionally, the average $NO_2$ concentration during working hours was 167.741 $\mu g/m^3$. Analysis of the working-hour data indicated that $PM_{2.5}$ posed a significantly greater health risk as its concentrations exceeded the recommended daily limits by a considerable margin. Regarding the main factors affecting construction dust and $NO_2$ concentrations, the results demonstrated the lack of a significant correlation with individual meteorological factors, despite minimal variations in these factors throughout the study. Given the weak correlation between PM and $NO_2$ concentrations and meteorological parameters, both multiple linear regression (MLR) and artificial neural network (ANN) models were employed for predictive purposes. The ANN model exhibited a better accordance with the measured air pollutant concentrations compared to the MLR model.

In the future, the research will concentrate on examining the influence of work dynamics (such as the number of construction equipment) on air quality. This investigation will consider meteorological parameters and predict the impact of various construction activities on air quality at the construction site.

**Author Contributions:** Conceptualization, S.M.K. and L.M.; methodology, Z.B.; software, S.M.K. and O.K.; validation, S.M.K. and L.M.; formal analysis, S.K.; investigation, S.M.K.; resources, Z.B.; data curation, S.K.; writing—original draft preparation, L.M.; writing—review and editing, Z.B.; visualization, L.M.; supervision, S.M.K.; project administration, S.K.; funding acquisition, Z.B. All authors have read and agreed to the published version of the manuscript.

**Funding:** This research was funded by Union-Nikola Tesla University, grant number 231/1.

**Institutional Review Board Statement:** Not applicable.

**Informed Consent Statement:** Not applicable.

**Data Availability Statement:** Models or code that support the findings of this study are available from the corresponding author upon reasonable request.

**Conflicts of Interest:** The authors declare no conflict of interest.

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
