# Peer review of "Study of Particular Air Quality and Meteorological Parameters at a Construction Site"

_atmosphere, doi:10.3390/atmos14081267_

Round 1
Reviewer 1 Report
The paper focus on relationship between particular air quality and meteorological parameters at a construction site. Authors implemented an IoT-based system framework that integrated a distributed sensor network to collect real-time data from the construction site on the concentration of NO2 and particulate matter (PM2.5 and PM10), as well as meteorological parameters such as wind speed, wind direction, humidity, pressure, and temperature. Multiple linear regression (MLR) and Artificial Neural Network (ANN) for PM2.5, PM10 and NO2 concentrations prediction in air that used meteorological parameters as predictors was applied. However, the manuscript has some major academic problems.
1. The descriptions in the lines 315-317 are contradictory. PM10 pollution comes from the south, southwest and west, dominantly (Figure 10b).The location of the excavation zone is west of the measuring station, that is consistent with the result.”
If three pollutants comes from the south, southwest and west, the real-time data on the air pollutant concentration collected cannot all be attributed to the earthworks.
2. Why do the two prediction models not take into account the influence of the construction activity intensity? As descripted in the article lines 287-297, construction activities directly contribute to the generation of construction dust, making them the primary driver of construction dust levels.
3. The study was conducted at a construction site in Belgrade over a 15-day period , and the measurement station recorded measurements every 5 minutes. This means that there were over 4000 experimental data available throughout the study period. However, the data were only 1,800(in line 326, A set of 1492 data was used, since not all of the expected 1800 data were valid.)There are huge differences in the amount of experimental data.
The quality of English language is good.
Reviewer 2 Report
Comments to Authors:
Abstract: Well written, with a short introduction about the research background and objective. However, throughout the article, please be careful to use the short abbreviation for air pollutants. For instance, it should be PM2.5 but not PM2.5. Please modify accordingly.
Introduction: Minor revision is necessary. Highlight the air pollution too. the current topic you are addressing is more to air quality/air pollution instead of climate change. Although air pollutants are the drivers for climate change, however it will be better to discuss air quality conditions. Highlight about the COVID-19 issues. Mentioning about the insufficient monitoring stations in developing countries, causing the detailed phenomenon were not reported. Some relevant studies which should be included.
(iii) Materials and methodology: Include a better mapping. The current resolution is too coarse. Also, provide a better illustration of the construction site. Utilize some real iconic to represent the buildings.
(iv) Include a research flow chart.
(v) Draw the input and output parameters for the models.
(vi) the sentence (minimum, 1st quartile, median and mean 245 value, 3rd quartile, and maximum, as well as outliers are shown) is not necessary.
(vii) I do not see the IoT part in the discussion. Better to have pictures illustrating the monitoring stations.
(viii) Figures for the results of meteorological and air quality can be combined. Quite weak in current section.
(ix) Change Table 1 into a heat map for the Correlation Analysis.
(x) Discussion is lacking in this study. So what are the optimizing methods used in the study? Comparison among performance? Validation of the IoT sensors? Some relevant literature which can be included to enhance the discussion are: (i)Quantification of COVID-19 impacts on NO2 and O3: Systematic model selection and hyperparameter optimization on AI-based meteorological-normalization methods
(xi). Conclusion section seems to be a repetition of the results section. Huge modifications are required. Please provide insights into this study and what can be further done in the future.
(xii). Implications for future research may also be included in the conclusion at the end.
Moderate editing of English language required
Round 2
Reviewer 1 Report
1. It would be even better to supplement the contribution of off-site factors near the construction site on the real-time data on the air pollutant concentration collected.
2. Please supplement the quantization data of the intensity of the construction activities like as earthworks during the entire measurement period.
Author Response
Dear Reviewer,
We are submitting a second corrected version of our manuscript. All the changes made in the manuscript are marked in yellow. All suggestions are taken into account. We are grateful for new suggestions that raised the quality of our manuscript. We sincerely hope that our answers are in accordance with the criteria and the quality of journal.
Sincerely,
Author and co-authors
Reviewer 1
1. It would be even better to supplement the contribution of off-site factors near the construction site on the real-time data on the air pollutant concentration collected.
Answer: Ln 184-187
2. Please supplement the quantization data of the intensity of the construction activities like as earthworks during the entire measurement period.
Answer: Ln 179-181
Reviewer 2 Report
The authors have substantially addressed my comments in the first review. There are still some minor comments to be addressed:
(i) Line 457-460: impact of work dynamics (number of working machines) on air quality, taking into account meteorological parameters, as well as the prediction of the impact of other construction activities, except earthworks, on air quality at the construction site.
What do you mean by the working machines?
I think you need not to mention except earthworks
(ii) Maybe proofreading is necessary to enhance the readability of the text.
(iii) The COVID-19 pandemic has had both positive and negative impacts on air pollution. The COVID-19 pandemic has revealed both the short-term improvements in air quality during lockdowns and the challenges in managing medical waste that indirectly affects air pollution. During the lockdowns and restrictions imposed to control the spread of the virus, there was a noticeable reduction in economic activities and transportation, leading to a temporary improvement in air quality...
Properly citation on Spatiotemporal impact of COVID-19 on Taiwan air quality in the absence of a lockdown: Influence of urban public transportation use and meteorological conditions maybe helpful to allow the readers to find the original source too.
Minor editing of English language required
Author Response
Dear Reviewer,
We are submitting a corrected version of our manuscript. All the changes made in the manuscript are marked in yellow. All suggestions are taken into account. We are grateful for new suggestions that raised the quality of our manuscript. We sincerely hope that our answers are in accordance with the criteria and the quality of journal.
Sincerely,
Author and co-authors
Reviewer 2
(i) Line 457-460: impact of work dynamics (number of working machines) on air quality, taking into account meteorological parameters, as well as the prediction of the impact of other construction activities, except earthworks, on air quality at the construction site.
What do you mean by the working machines?
Answer: „Construction equipment“ or „construction machinery“ (excavator, backhoe, bulldozer)
I think you need not to mention except earthworks
Answer: Removed
(ii) Maybe proofreading is necessary to enhance the readability of the text.
Answer: Done
(iii) The COVID-19 pandemic has had both positive and negative impacts on air pollution. The COVID-19 pandemic has revealed both the short-term improvements in air quality during lockdowns and the challenges in managing medical waste that indirectly affects air pollution. During the lockdowns and restrictions imposed to control the spread of the virus, there was a noticeable reduction in economic activities and transportation, leading to a temporary improvement in air quality...
Properly citation on Spatiotemporal impact of COVID-19 on Taiwan air quality in the absence of a lockdown: Influence of urban public transportation use and meteorological conditions maybe helpful to allow the readers to find the original source too.
Answer: Done
Wong, Y. J., Shiu, H. Y., Chang J. H. H., Ooi, M. C. G., Li, H. H., Homma, R., Shimizu, Y., Chiueh, P. T., Maneechot, L., Sulaiman, N. M. N. (2022) Spatiotemporal impact of COVID-19 on Taiwan air quality in the absence of a lockdown: Influence of urban public transportation use and meteorological conditions, Journal of Cleaner Production,Volume 365, 132893, https://doi.org/10.1016/j.jclepro.2022.132893